# Self-binding of one-dimensional fermionic mixtures with zero-range interspecies attraction

**Jules Givois⋆, Andrea Tononi, and Dmitry S. Petrov**

Université Paris-Saclay, CNRS, LPTMS, 91405 Orsay, France

⋆ jules.givois@universite-paris-saclay.fr

## Abstract

For sufficiently large mass ratios the attractive exchange force caused by a single light atom interacting with a few heavy identical fermions can overcome their Fermi degeneracy pressure and bind them into an $N + 1$ cluster. Here, by using a mean-field approach valid for large $N$, we find that $N + 1$ clusters can attract each other and form a self-bound charge density wave, the properties of which we fully characterize. Our work shows that there are no fundamental obstacles for having self-bound states in fermionic mixtures with zero-range interactions.



## 1 Introduction

According to our current understanding of Nature, big composite self-bound objects (nuclei, atoms, molecules, liquids, solids, etc.) are formed due to attractive finite-range forces originating from exchanges of bosons (gluons, photons, etc.) However, one of the archetypal fermionic models, the two-component mass-balanced Fermi gas with zero-range attraction [1], exhibits

no self-binding. Increasing the attraction in such a gas leads to the formation of dimers consisting of fermions of different sort. The dimers repel each other in any space dimension because of the Pauli exclusion principle for the constituent fermions [2–9]. The Pauli "repulsion" is also believed to be the main mechanism preventing binding of four neutrons, although the topic remains controversial since internucleon interactions are not zero range [10].

Introducing mass imbalance into the model can lead to binding of mesoscopic clusters of type $N+1$, where the exchange force of a single light atom overcomes the degeneracy pressure of a small Fermi sea of $N$ heavy atoms. Such clusters with $N$ up to 5 have been studied by exact few-body techniques in all dimensions [11–18] and we have recently developed their mean-field theory in one dimension valid for large $N$ [19].

Can a two-component Fermi mixture with zero-range interactions become self-bound in the thermodynamic limit? A good starting point to answer this question is to understand whether two clusters of the type $N+1$ can stick together. This problem is nontrivial; the light atoms should be sufficiently light to ensure attraction for the heavies, but, on the other hand, their own degeneracy pressure (inversely proportional to the light mass) can hinder binding. No evidence of such binding has been reported. Here we can cite rather extensive studies of the fermionic 2+2 system and the dimer-dimer scattering problem in the mass-imbalanced case [20–26]. Although not fully comprehensive (i.e., not all dimensions and possible mass ratios covered), these studies are consistent with the scenario that the 2+2 fermionic system is either unbound or breaks into two repulsive dimers or into a heavy-heavy-light trimer plus a free light atom when the trimer gets below the two-dimer threshold. Naidon and co-workers [27, 28] estimated that three-dimensional 2+1 trimers repel each other.

In this article we show that in one dimension $N+1$ clusters can arrange themselves into a self-bound configuration, at least, for sufficiently large $N$. To this end we use the mean-field theory based on the Thomas-Fermi approximation for the heavy atoms, valid in the limit $N \gg 1$ [19]. In this case, the system behavior is governed by a single parameter $\alpha = (\pi^2/3)N^3 m/M$. We find that two clusters bind for $2.3 < \alpha < 9.4$. Interestingly, instead of merging, they stay at a finite distance from each other and keep a double-peak density profile (see the gray dotted curve in Fig. 2). Below $\alpha = 2.3$ this state becomes metastable and the true ground state corresponds to two $N+1$ clusters at infinite separation. Our calculations show that three or more clusters can form a self-bound charge density wave with one light atom per period. We describe bulk properties of these states by a fully analytic weak-modulation theory, the Peierls instability [29] emerging as a complementary explanation of the modulation.

## 2 Model and $N+1$ cluster solution

We address the problem of $N_h$ fermions of mass $M$ interacting with $N_l$ fermions of mass $m$ through the mean-field density functional

$$\Omega = \int dx \Big\{ \sum_{i=1}^{N_l} \big[|\partial_x \phi_i(x)|^2/2m + gn(x)|\phi_i(x)|^2\big] + \pi^2 n(x)^3/6M - \sum_{i=1}^{N_l} \epsilon_i |\phi_i(x)|^2 - \mu n(x) \Big\}, \quad (1)$$

where $g < 0$ is the heavy-light interaction constant, $\phi_i(x)$ are the wave functions of the light atoms, and $n(x)$ is the density profile of the heavy atoms. The term $\propto n^3(x)$ is the kinetic energy of an ideal Fermi gas taken in the Thomas-Fermi local-density approximation[1] [19]. The model (1) thus requires weak interactions ($a \gg \lambda_h, \lambda_l$) and $n(x)$ slowly varying on

---

[1]The model (1) is also applicable when one or both fermionic species are replaced by impenetrable bosons. One can also write it with a different equation of state for heavy atoms. In particular, self-bound Bose-Fermi liquids have been studied by using Eq. (1) with $\pi^2 n^3/6M$ replaced by the bosonic mean-field energy density $|\partial_x \sqrt{n}|^2/2M + g_{BB}n^2/2$ in Ref. [30].

the scale $\sim \lambda_h$. Here, $a = -(m+M)/(mMg)$ is the scattering length and $\lambda_h \sim 1/n$ and $\lambda_l \sim |\phi_i/\partial_x\phi_i|$ are, respectively, the typical de Broglie wave lengths of the heavy and light atoms. The first line in Eq. (1) defines the total energy $E$, which we seek to minimize subject to constraints $\int \phi_i^*(x)\phi_j(x)dx = \delta_{ij}$[2] and $\int n(x)dx = N_h$. The normalization constraints are taken into account by introducing Lagrange multipliers $\epsilon_i$ and $\mu$.

One can show that up to an overall scaling factor, the behavior of the system satisfying Eq. (1) is governed by two dimensionless parameters. We choose the first to be $N_l$ and the second to be $\alpha = (\pi^2/3)N^3 m/M$, where $N = N_h/N_l$. Indeed, introducing the characteristic size $\lambda = 1/(2m|g|N)$, new coordinate $u = x/\lambda$, and rescaling $\phi_i(x) = \tilde{\phi}_i(u)/\sqrt{\lambda}$, $n(x) = N\tilde{n}(u)/\lambda$, Eq. (1) reduces to

$$\frac{\Omega}{2mg^2N^2} = \int du \left\{ \sum_{i=1}^{N_l} \left[ |\partial_u\tilde{\phi}_i(u)|^2 - \tilde{n}(u)|\tilde{\phi}_i(u)|^2 \right] + \alpha\tilde{n}^3(u) - \sum_{i=1}^{N_l} \tilde{\epsilon}_i|\tilde{\phi}_i(u)|^2 - \tilde{\mu}\tilde{n}(u) \right\}, \quad (2)$$

where $\tilde{\mu} = 2mN\mu\lambda^2 < 0$, $\tilde{\epsilon}_i = 2m\epsilon_i\lambda^2$ and the normalization constraints are now $\int \tilde{\phi}_i^*(u)\tilde{\phi}_j(u)du = \delta_{ij}$ and $\int \tilde{n}(u)du = N_l$.

We minimize $\Omega$ imposing that the variational derivatives of Eq. (2) with respect to $\tilde{\phi}_i^*$ and $\tilde{n}$ vanish. These conditions, respectively, lead to the equations

$$-\partial_u^2\tilde{\phi}_i(u) - \tilde{n}(u)\tilde{\phi}_i(u) = \tilde{\epsilon}_i\tilde{\phi}_i(u), \quad (3)$$

$$\tilde{n}(u) = (3\alpha)^{-1/2}\text{Re}\sqrt{\sum_{i=1}^{N_l} |\tilde{\phi}_i(u)|^2 + \tilde{\mu}}. \quad (4)$$

The energy of the system then equals

$$\frac{E}{2mg^2N^2} = \sum_{i=1}^{N_l} \tilde{\epsilon}_i + \alpha\int \tilde{n}^3(u)du. \quad (5)$$

Let us briefly summarize the main results obtained for the case $N_l = 1$ [19]. The $N+1$ cluster exists for $\alpha < 12$ and we show its energy, denoted by $E_{N+1}^{N_l=1}$, as a function of $\alpha$ in the inset of Fig. 1. In the limit $\alpha \to 0$ the heavy atoms are much more localized than the light one, and the system can be described as a light atom bound by a point-like potential $Ng\delta(x)$ [we have $E_{N+1}^{N_l=1}/(2mg^2N^2) \to -1/4$]. With increasing $\alpha$ the heavy atoms get more freedom and their chemical potential grows till it reaches $\mu = 0$ at $\alpha = 12$. This corresponds to the right endpoints of the red dashed curves in Fig. 1 [here $E_{N+1}^{N_l=1}/(2mg^2N^2) = -1/60$]. Beyond this point the droplet cannot accomodate more heavy atoms. The validity of the model (2) is verified by the following arguments. Note that Eqs. (3) and (4) are dimensionless and, for $\alpha \sim 1$ and $N_l \sim 1$, the spatial extent of the cluster is of order $\lambda$ (in original units). The condition of the slowly varying density can be written as $|\partial_x n(x)| \ll n^2$ and translates to $N \gg 1$. This inequality is equivalent to $(m/M)^{1/3} \ll 1$ since we are mainly interested in $\alpha \sim 1$. We thus have $a \approx -1/mg$, or, in rescaled units, $\tilde{a} = a/\lambda \approx -2N$, which is much larger than $\tilde{\lambda}_h \sim 1/N$ and $\tilde{\lambda}_l \sim 1$, ensuring weak interactions.

## 3 Binding of two or more $N+1$ clusters

In the case $N_l = 2$ we solve Eqs. (3) and (4) iteratively. Namely, we diagonalize Eq. (3) assuming a certain initial $\tilde{n}$, substitute the obtained $\tilde{\phi}_i$ into Eq. (4), tune $\tilde{\mu}$ to satisfy the

---

[2]These constraints originate from the fact that the light-atom part of Eq. (1) is the Hartree-Fock variational energy built on the $N_l \times N_l$ Slater determinant of the orbitals $\phi_i(x)$.

normalization constraint for $\tilde{n}$, and repeat the procedure till convergence. In this manner, we obtain three types of solutions. The first type is two isolated $N + 1$ clusters, the second is a bound state of two $N + 1$ clusters, and the third is a $2N + 1$ cluster plus a free light atom. The first type is the ground state of the $2N + 2$ system for $0.16 < \alpha < 2.3$.

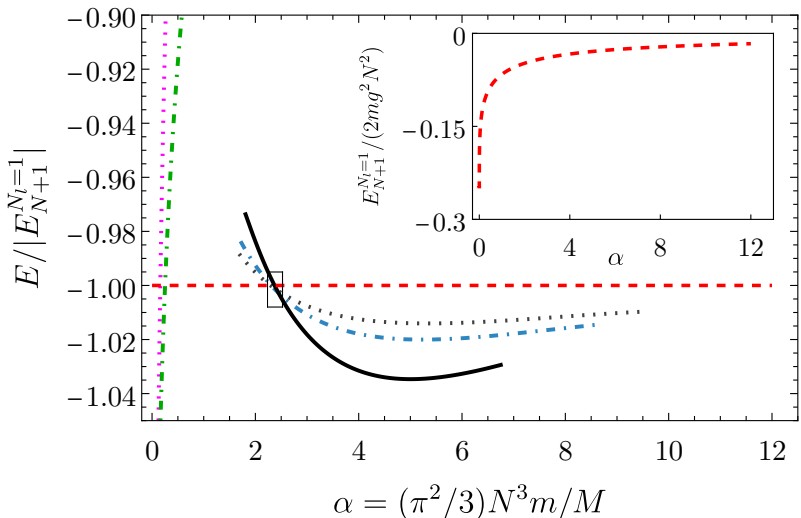

Figure 1: Energies per cluster $E_{N+1}^{N_l=1}$ (red dashed), $E_{N+1}^{N_l=2}$ (gray dotted), $E_{N+1}^{N_l=3}$ (blue dash-dotted), and $E_{N+1}^{N_l=\infty}$ (black solid) in units of the energy of an isolated $N + 1$ cluster $E_{N+1}^{N_l=1}$ shown in the inset. For small $\alpha$, $N_l$ isolated $N + 1$ clusters can lose one of their light atoms and rearrange into $N_l - 1$ isolated clusters with larger $N$. The final-to-initial energy ratio (with minus sign) is shown for $N_l = 2$ (magenta dotted) and $N_l = 3$ (green dash-dotted). The crossing region (black frame) will be shown in more detail in Fig. 3.

The second type is realized in the region $\alpha > 1.6$. The corresponding energy per cluster (we denote it by $E_{N+1}^{N_l=2}$) is shown in Fig. 1 as the gray dotted curve. For our purposes it is convenient to normalize all energies in Fig. 1 to the energy of an isolated $N + 1$ cluster, shown as the horizontal red dashed line. For $\alpha > 2.3$ the bound-state configuration is the ground state, and in the region $1.6 < \alpha < 2.3$ it is only dynamically stable (thermodynamically it prefers to break into isolated clusters). This state is characterized by a density profile with two maxima separated by a finite distance (see Fig. 2). Below $\alpha = 1.6$ the metastable state disappears and the clusters unbind. To qualitatively understand this phenomenon, we have performed a variational analysis by taking $\tilde{n}(u)$ as a sum of two Gaussians with variable width $\tilde{\sigma}$, placed at distance $\tilde{\xi}$ from each other. The minimization of Eq. (5) with respect to $\tilde{\sigma}$ then gives the energy as a function of $\tilde{\xi}$. The curves $E(\tilde{\xi})$ obtained in this manner can feature a (meta)stable minimum at finite $\tilde{\xi}$. They are very similar to what we obtain for infinite $N_l$ (see Sec. 4 and Fig. 4). The left panel in Fig. 4 corresponds to the critical $\alpha$ where the metastable minimum disappears. This is the point where the minimum and maximum of $E(\tilde{\xi})$ merge, creating nonanalytical singularities in both the optimal $\tilde{\xi}$ and the energy. The same nonanalytic behavior is observed in our exact (not variational) numerics. In fact, the critical points in Fig. 1 are determined by gradually decreasing $\alpha$ and monitoring the distance between the peaks and the energy of the systems.

We find that the curves $E(\tilde{\xi})$ obtained by the variational procedure are characterized by a repulsive tail at large $\tilde{\xi}$. That the clusters repel each other at large separations is due to the exchange of the identical light fermions. The mechanism can be understood in the Born-Oppenheimer approximation and is similar to the long-distance repulsion between two het-

eronuclear mass-imbalanced dimers [23]. The minimum at finite $\tilde{\xi}$ appears because heavy atoms distribute their density in an optimal manner to provide binding in spite of the degeneracy pressure. This phenomenon, which is the main result of this work, is not obvious and rather subtle (note relatively low binding energies). As we increase $\alpha$ beyond 9.4, similarly to the case $N_l = 1$, the chemical potential $\mu$ crosses zero and becomes positive. This simply means that, in free space, the cluster will eject the excess of heavy atoms, effectively decreasing $\alpha$ to subcritical values.

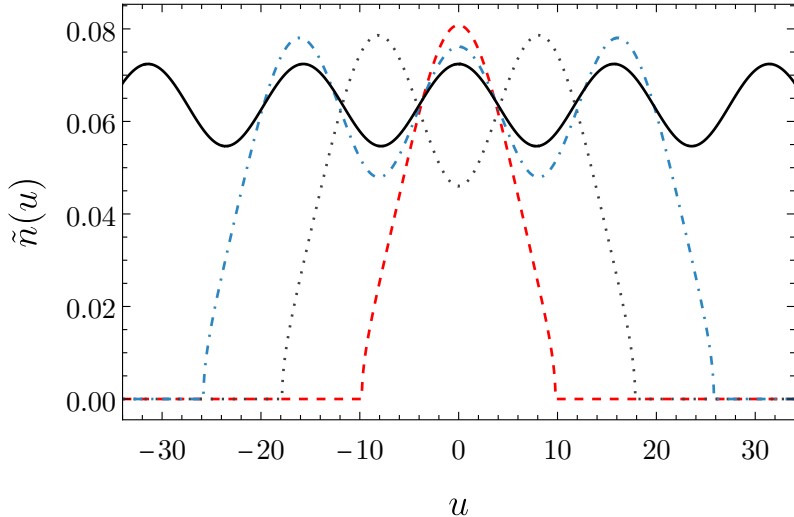

Figure 2: Heavy-atom density profiles for a single $N+1$ cluster (red dashed) and for chains of length $N_l = 2$ (gray dotted), $N_l = 3$ (blue dash-dotted), and $N_l = \infty$ (black solid) for $\alpha = 4.45$.

Still speaking about solutions of the second type we find that the phenomenon of self-binding persists for $N_l > 2$. Solving Eqs. (3) and (4) for $N_l$ up to 5 we observe that clusters tend to form a regular chain or polymer, with the number of density peaks equal to $N_l$. To avoid cluttering in Figs. 1 and 2 we show only the case $N_l = 3$ (dash-dotted) in addition to $N_l = 1, 2$ already discussed. The three-cluster bound state shows qualitatively the same behavior as the two-cluster state. In brief, it is stable above and metastable below $\alpha = 2.4$. Below $\alpha = 2.2$ the system energetically prefers to break into three isolated $N+1$ clusters. An interesting feature of this system[3] is that in the region $2.2 < \alpha < 2.4$ (more precisely $2.21 < \alpha < 2.38$) its true ground state is an $N_l = 2$ chain with $N_h = 2N'$ plus a single $(3N-2N')+1$ cluster, where $N'$ is determined by the minimization of $2E_{N'+1}^{N_l=2} + E_{(3N-2N')+1}^{N_l=1}$.

We observe similar behavior for higher $N_l$. The curves corresponding to the binding energies (per cluster) of the chains with different $N_l$ bundle together in the region $\alpha \approx 2.4 \pm 0.1$. In Fig. 3 we plot the results for $N_l = 1, 2, 3, 4, 5$, and $\infty$. The determination of the ground state for a given $N_l$ in this region is complicated since a longer chain can break into shorter chains with generally different $\alpha$. We note, however, that in this region all partitions of a chain are almost degenerate and correspond to (meta)stable states separated by energy barriers similar to the one shown in the middle panel of Fig. 4. For larger $\alpha$, sufficiently far from the crossing region, clusters do prefer to merge into a single chain since longer chains feature higher binding energy per cluster.

The third type of solutions of Eqs. (3) and (4) realizes for small $\alpha$ when $N_l$ isolated $N+1$ clusters eject one light atom forming $N_l - 1$ isolated $N'+1$ clusters with $N' = NN_l/(N_l-1)$, the new configuration becoming energetically favorable for $(N_l-1)E_{N'+1}^{N_l=1} < N_l E_{N+1}^{N_l=1}$. In Fig. 1

---

[3]We thank one of the reviewers of the first version of the manuscript for bringing this point to our attention.

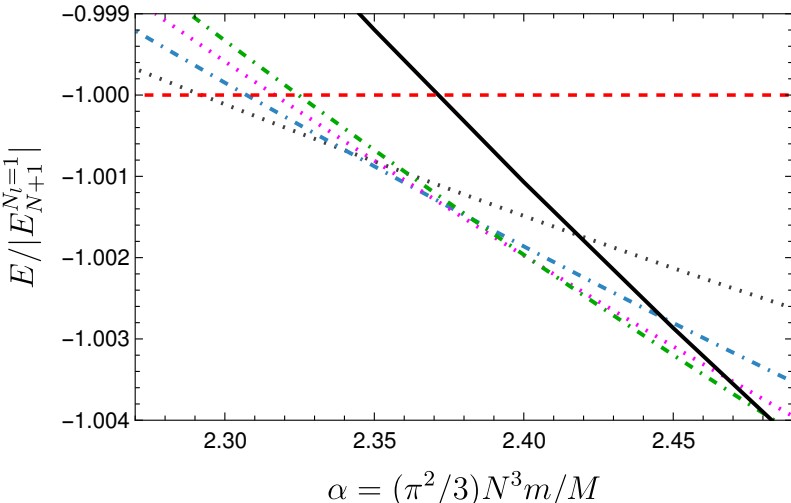

Figure 3: Zoom into the framed region in Fig. 1. Here, in addition, we show the energies for $N_l = 4$ (magenta dotted) and $N_l = 5$ (green dash-dotted). The curves show that for a given $N_l$ the longest possible chain or $N_l$ isolated clusters are not the only possible ground states. For instance, for $N_l = 3$, the energetically optimal configuration can be a cluster with one light atom isolated from a cluster with 2 light atoms (both clusters having generally different values of $\alpha$, see text).

we show the quantity $(N_l - 1)E_{N'+1}^{N_l=1}/N_l|E_{N+1}^{N_l=1}|$ for $N_l = 2$ (magenta dotted) and $N_l = 3$ (green dash-dotted). These curves cross the line -1 at $\alpha = 0.16$ and $\alpha = 0.25$, respectively. This critical value of $\alpha$ grows with $N_l$ reaching 0.47 in the thermodynamic limit, where it is obtained from the condition $\partial[E_{N+1}^{N_l=1}/N]/\partial N = 0$. In principle, starting from a long chain of isolated clusters and trying to follow its ground state by decreasing $\alpha$ below 0.47, the system will lose light atoms one by one till it eventually ends up in the state where all $N_h$ heavy atoms are bound by a single remaining light atom. We should note, however, that these transitions are associated with a global redistribution of the heavy particles such that the system will likely get stuck in metastable states with "wrong" $N_l$. This is because these transitions are not associated with $\max[\epsilon_i]$ crossing zero (isolated clusters individually never lose their light atoms).

## 4 Infinite chain analysis

We now go back to the second type of solutions and discuss bulk properties of self-bound chains in more detail. We assume that $\tilde{n}(u)$ is periodic with the modulation length $\tilde{\xi}$ and that light atoms are filling the first Brillouin zone of the lattice (we have one light atom per modulation length). We aim to calculate the energy per cluster, which we denote $E_{N+1}^{N_l=\infty}$, as a function of $\tilde{\xi}$. In principle, Eqs. (3-5) are suitable for the task. In this case $\tilde{\phi}_i$ become Bloch functions and $i$ is the real Bloch wave vector in the first Brillouin zone, i.e., $i \in (-\pi/\tilde{\xi}, \pi/\tilde{\xi}]$. It is however convenient to rescale the coordinate again, introducing $\bar{u} = u/\tilde{\xi}$. Making related changes and rescalings (we mark new rescaled quantities by a bar), we arrive at the following formulation of the problem. The energy per cluster is given by

$$\frac{E_{N+1}^{N_l=\infty}(\tilde{\xi})}{2mg^2N^2} = \frac{1}{\tilde{\xi}^2}\left[\int_{-\pi}^{\pi}\bar{\epsilon}_p\frac{dp}{2\pi} + \alpha\int_0^1\bar{n}^3(\bar{u})d\bar{u}\right], \tag{6}$$

where the spectrum $\bar{\epsilon}_p$ is determined by the equation

$$(-i\partial_{\bar{u}} + p)^2 \chi_p(\bar{u}) - \tilde{\xi}\bar{n}(\bar{u})\chi_p(\bar{u}) = \bar{\epsilon}_p \chi_p(\bar{u}), \tag{7}$$

with the periodic boundary condition $\chi_p(0) = \chi_p(1)$. The function $\chi_p$ is the periodic part of the Bloch wave function corresponding to the wave vector $p \in (-\pi, \pi]$. The density $\bar{n}$ is given by

$$\bar{n}(\bar{u}) = (3\alpha)^{-1/2}\text{Re}\sqrt{\tilde{\xi}\int_{-\pi}^{\pi}|\chi_p(\bar{u})|^2\frac{dp}{2\pi} + \bar{\mu}}, \tag{8}$$

where $\bar{\mu} = \tilde{\mu}\tilde{\xi}^2$, and the normalization conditions read $\int_0^1 |\chi_p(\bar{u})|^2 d\bar{u} = 1$ and $\int_0^1 \bar{n}(\bar{u})d\bar{u} = 1$.

We solve Eqs. (7) and (8) iteratively, calculating $E_{N+1}^{N_l=\infty}(\tilde{\xi})$ for various $\alpha$. A few examples of these curves are shown in Fig. 4. We find that there is always a local minimum at $\tilde{\xi} = \infty$ corresponding to isolated noninteracting clusters $[E_{N+1}^{N_l=\infty}(\tilde{\xi} \to \infty) = E_{N+1}^{N_l=1}]$. We also see that $E_{N+1}^{N_l=\infty}(\tilde{\xi})$ decreases with $\tilde{\xi}$ for small $\tilde{\xi}$. This is understandable as we are dealing with two Fermi seas at high densities $\propto 1/\tilde{\xi}$ where the interaction is asymptotically negligible.

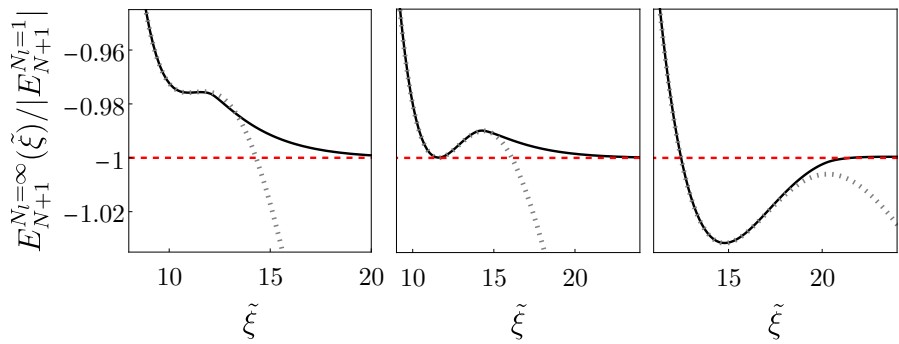

Figure 4: Energies per cluster in an infinite chain as a function of the distance between the clusters for $\alpha = 1.85$ (left), $\alpha = 2.4$ (middle) and $\alpha = 4.0$ (right). The black solid curves are obtained by numerically solving Eqs. (7) and (8). The gray dotted curves are predictions of the weak-modulation theory Eq. (15).

The function $E_{N+1}^{N_l=\infty}(\tilde{\xi})$ has a local (for $1.8 < \alpha < 2.4$) or global (for $\alpha > 2.4$) minimum at finite $\tilde{\xi}$. This minimum can correspond to the self-bound solution, as this is the point of zero pressure. The energy per cluster in this state is shown in the main panel of Fig. 1 (black solid curve). However, we should specify that the minimum of $E_{N+1}^{N_l=\infty}(\tilde{\xi})$ does not necessarily mean that this state is self-bound in free space. We have yet to check the conditions $\mu < 0$ (otherwise the chain will lose heavy atoms) and $\max[\bar{\epsilon}_p] = \bar{\epsilon}_\pi < 0$ (otherwise it will lose light atoms). In fact, the right end point of the black solid curve in Fig. 1 corresponds to $\bar{\mu} = 0$. The behavior of the self-bound chain near this point is thus similar to what happens in the cases $N_l = 1$ and $N_l = 2$ already discussed. By contrast, the left end point does correspond to the disappearance of the minimum (see left panel in Fig. 4). There, the chain breaks into free $N + 1$ clusters. We should mention that by assuming one light atom per period we disregard possible breaking of a long chain into smaller clusters sufficiently close to $\alpha = 2.4$ as we have pointed out in Sec. 3.

Although there is no apparent small parameter that allows us to solve Eqs. (7) and (8) perturbatively, the assumption of weak modulation, which provides a completely analytic description of the system, turns out to work extremely well for all values of $\alpha$ and $\tilde{\xi}$ relevant for

analyzing the self-bound regime. The weak-modulation theory is based on the ansatz

$$\bar{n}(\bar{u}) = 1 + (\delta/\tilde{\xi})\cos(2\pi\bar{u}), \qquad (9)$$

where $\delta$ is assumed to be small. One then calculates $E_{N+1}^{N_l=\infty}(\tilde{\xi})$ given by Eq. (6) up to terms $\propto \delta^2$, and minimizes it with respect to $\delta$.

We calculate the spectrum $\bar{\epsilon}_p$ following the standard weak-modulation approach [31]. Namely, substituting the expansion $\chi_p = \sum_{j=-\infty}^{\infty} \beta_j e^{i2\pi j\bar{u}}$ into Eq. (7), we get the set of equations

$$[(p+2\pi j)^2 - \tilde{\xi} - \bar{\epsilon}_p]\beta_j - \delta(\beta_{j-1}+\beta_{j+1})/2 = 0, \qquad (10)$$

for all integer $j$. Since $\bar{\epsilon}_p = \bar{\epsilon}_{-p}$ we can consider only the positive half of the first Brillouin zone, i.e., $0 < p < \pi$. One can then check that the solution of Eqs. (10) for the lowest band is characterized by the following hierarchy. The coefficients $\beta_0$ and $\beta_{-1}$ are of order one or smaller, $\beta_1$ and $\beta_{-2}$ are $\sim \delta$ or smaller, $\beta_2$ and $\beta_{-3}$ are $\sim \delta^2$, etc. Therefore, up to the second order in $\delta$ we can write $\beta_1 = (\delta/2)\beta_0/[(p+2\pi)^2 - p^2]$, where we use Eq. (10) with $j = 1$ and neglect the small difference (at most $\propto \delta$) between $\bar{\epsilon}_p$ and the unpertubed energy $p^2 - \tilde{\xi}$. In a similar way, Eq. (10) with $j = -2$ gives $\beta_{-2} = (\delta/2)\beta_{-1}/[(p-4\pi)^2 - p^2]$. Substituting these $\beta_1$ and $\beta_{-2}$ into Eq. (10) with $j = 0$ and $j = -1$ we obtain

$$\{p^2 - \tilde{\xi} - \bar{\epsilon}_p - (\delta/2)^2/[(p+2\pi)^2 - p^2]\}\beta_0 - (\delta/2)\beta_{-1} = 0, \qquad (11)$$

$$\{(p-2\pi j)^2 - \tilde{\xi} - \bar{\epsilon}_p - (\delta/2)^2/[(p-4\pi)^2 - p^2]\}\beta_{-1} - (\delta/2)\beta_0 = 0. \qquad (12)$$

Solving this linear system gives the spectrum $\bar{\epsilon}_p$ with the desired accuracy.

To integrate $\bar{\epsilon}_p$ we divide the $p > 0$ part of the Brillouin zone into two regions. The first is $0 < p < \pi - \sqrt{\delta}$. Here we just use the Taylor expansion of $\bar{\epsilon}_p$ up to terms $\propto \delta^2$. In the remaining interval $\pi - \sqrt{\delta} < p < \pi$ we cannot Taylor expand $\bar{\epsilon}_p$ as this would lead to a divergent integral. However, replacing $p$ by $\pi$ in the terms proportional to $\delta^2$ in Eqs. (11) and (12) (one can check that this is a legal approximation in the considered integration interval) gives $\bar{\epsilon}_p$ in the form suitable for analytic integration. The result is

$$\int_{-\pi}^{\pi} \bar{\epsilon}_p \frac{dp}{2\pi} = -\tilde{\xi} + \frac{\pi^2}{3} + \frac{\delta^2}{16\pi^2}\ln\frac{\delta}{16\pi^2\sqrt{e}}. \qquad (13)$$

Finally, using $\int_0^1 \bar{n}^3(\bar{u})d\bar{u} = 1 + (3/2)\delta^2/\tilde{\xi}^2$, the minimization of Eq. (6) with respect to $\delta$ gives

$$\delta = 16\pi^2 e^{-24\pi^2\alpha/\tilde{\xi}^2}, \qquad (14)$$

and

$$\frac{E_{N+1}^{N_l=\infty}(\tilde{\xi})}{2mg^2N^2} = \frac{1}{\tilde{\xi}^2}\left(\alpha - \tilde{\xi} + \frac{\pi^2}{3} - 8\pi^2 e^{-48\pi^2\alpha/\tilde{\xi}^2}\right). \qquad (15)$$

To complete the theory, we note that the chemical potential can be determined by raising Eq. (8) to the second power and by integrating the result over $\bar{u}$. We obtain $\bar{\mu} = 3\alpha[1 + \delta^2/(2\tilde{\xi}^2)] - \tilde{\xi}$.

In Fig. 4 we show $E_{N+1}^{N_l=\infty}(\tilde{\xi})$ in units of the single-cluster energy $|E_{N+1}^{N_l=1}|$ as a function of the modulation period for $\alpha = 1.85$, 2.4 and 4.0. The black solid curves are determined by exactly solving Eqs. (7) and (8) and the gray dotted curves are given by Eq. (15). The weak-modulation approximation is very precise (much less than a percent deviation from the exact numerics) up to the minima for all considered $\alpha$. An appreciable difference can be seen only at large $\tilde{\xi}$, far from the minima.

# 5 Conclusion

In conclusion, we show that two $N+1$ clusters formed in a two-component fermionic mixture can attract each other by a peculiar potential with a minimum at a finite inter-cluster separation. This attraction persists in the thermodynamic limit such that the Fermi-Fermi mixture can become self-bound forming a polymer of $N+1$ clusters. One can also think of this state as a self-bound homogeneous liquid, undergone the Peierls charge density wave instability[4] [29]. Since our theory is valid for $N \gg 1$ [or $(M/m)^{1/3} \gg 1$] we cannot determine the smallest $N$ [or $M/m$] at which $N+1$ clusters bind. This problem should be tackled by other methods such as, for instance, exact diagonalization, quantum Monte-Carlo, or density matrix renormalization group. Our theory, taken at its face value, predicts binding of 4+1 clusters in the fermionic $^{173}$Yb-$^{6}$Li mixture ($M/m = 28.75$), which can in principle be checked in current experiments [32, 33].

**Funding information**   We acknowledge support from ANR Grant Droplets No. ANR-19-CE30-0003-02.

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
