# Peer review of "Self-binding of one-dimensional fermionic mixtures with zero-range interspecies attraction"

_SciPost Physics, doi:SciPost Phys. 14, 091 (2023)_

## Round 1 · Referee Report · Anonymous (Referee 2) · 2022-12-5

Strengths

1. The results are original and significant.
2. The theory is detailed, complete and valid.
3. The figures are useful.

Weaknesses

1. The conclusion could be more substantial (see below).
2. There are a number of minor grammatical errors, however these did not impede meaning or alter context.

Report

The study in this paper is based on the phenomenon whereby a single light atom can bind N heavy identical fermions into an N+1 cluster at sufficiently large mass ratios. In isolation, the Pauli exclusion principle (Fermi degeneracy pressure) would prevent the heavier atoms from forming a bound state. However, the lighter particle acts as a mediator that “glues” the (N+1)-body system together by way of an attractive exchange interaction.

Working in one dimension, the authors minimize the density functional to obtain the energy per cluster as a function of a dimensionless parameter “α”, for an increasing number of the light atoms. Here, “α” depends on the ratios of both the numbers and masses of the heavy and light particles. Their main result is that in a special regime of “α”, two such clusters can bind while remaining a finite distance apart — i.e., they maintain a double-peaked density profile. Subsequently, the authors observe this phenomenon of self-binding for higher numbers of clusters, which they show tend to form regular chains or polymers with the number of density peaks equal to the number of light atoms. They analyse the bulk properties of these self-bound chains (or charge density waves) in detail and evoke the Peierls instability to explain the observed density modulation.

The theoretical treatment is very detailed and thorough — there seems to be no missing information — and its testability and relevance to current experiments are also discussed. All aspects and features of the figures are clearly addressed and explained. There are several useful additions appearing in this version of the manuscript compared to the first version that is available on the arXiv. The results on self-binding have important implications for theoretical and experimental studies on mass- and population-imbalanced Fermi-Fermi mixtures, and I would recommend it for publication provided the authors address the following minor remarks.

Requested changes

1. Please add an appropriate reference to the second sentence of the last paragraph of the introduction ([19]?).
2. The conclusion seems a bit brief... I think it would be worth adding a sentence on your findings for only two N+1 clusters (to complement what you say about a polymer of these clusters in the thermodynamic limit). Also, you mention that the small-N problem should be tackled by other methods. Any suggestions of what a useful approach might be in this case?

---

## Round 1 · Referee Report · Anonymous (Referee 3) · 2023-1-6

Strengths

1. The results are original and timely
2. The work predicts regimes for the existence of self-bound polymers of clusters in an attractive Fermi-Fermi mixture with mass imbalance
3. The theoretical predictions should be testable by available experimental techniques

Weaknesses

1. The validity of mean-field theory is only guaranteed in the large-particle-number regime for the heavy species
2. Results on metastable regimes lack details (see report)

Report

The manuscript investigates the binding of clusters in a Fermi-Fermi mixture with mass imbalance in one spatial dimension by means of mean-field theory. Some of the results for an infinite modulated two-component fluid are obtained by analytical means, while other results for a finite number of light fermions are numerical. The results are interesting and the topic is very timely. The manuscript is well written I have no doubts in the validity of the main results. A limitation of the underlying mean-field theory, as is clearly stated in the manuscript, is that it cannot be expected to provide accurate results for finite particle numbers (of heavy atoms).

The results should definitely be published but I would like the authors to consider the following comments and questions:

Requested changes

1. In section 3 on page 4 a claim is made about the existence of metastable bound states of clusters for finite N_l, e.g. for 1.6 < alpha < 2.3 but it is unclear on which basis this conclusion is drawn as no evidence is presented besides the lines drawn in Fig. 1. Was this conclusion drawn from fully self-consistent numerical calculations, or from the variational analysis? Do we know that these solutions are metastable or might they be (dynamically) unstable solutions of the numerical self-consistent procedure? Why does the corresponding line (e.g. N_l = 2) in Fig. 1 not extend to the full interval of claimed metastability (It looks like the grey dotted line terminates at > 1.8)? Ideally I would like to see a stability analysis of the numerically found non-ground-state solutions, or at least a clarification about what is known about metastability and how the conclusions were reached.

2. The beginning of Sec. 3 on page 3 announces three types of solutions, but the third solution is only mentioned two pages later on page 5. To improve the readability, I would suggest to briefly summarise the character of the three solutions at the beginning of the section before discussing them in detail.

---

## Round 2 · Referee Report · Anonymous (Referee 1) · 2023-1-12

Report

I am very happy with the way in which the authors have responded to the comments in my first report. In my opinion, the revised manuscript meets all of the "general acceptance criteria" and at least one of the "expectations" for SciPost Physics. In brief, the article conclusively predicts regimes of existence for self-bound polymers of clusters in fermionic mixtures with zero-range attractive interactions. This is a new and significant result which can be immediately checked experimentally (in principle). Therefore, the article "details a groundbreaking theoretical/computational discovery", and I believe that it is now suitable for publication.

---

## Round 2 · Referee Report · Anonymous (Referee 2) · 2023-1-14

Report

When reading the authors' response I realised that I had misread the scale of Fig. 1: The grey dotted line ($\alpha$ regime for the existence of a bound 2N+2 system) indeed terminates at $\alpha = 1.6$.

The authors' response has clarified the basis for the results regarding the metastability of bound states, which had been a concern of mine. Metastability was concluded on the basis of the variational analysis, which shows a local energy minimum for the candidate solutions in the relevant regime.

I would like to note that, while reasonable, the metastable character is not fully proven by this analysis. As the variational analysis reduces the infinite number of degrees of freedom of the mean-field theory down to a single variable, here the peak separation $\tilde{\xi}$, it is still possible that what appears as a local minimum in the collective degree of freedom might be a saddle along an another, not-captured, coordinate direction. To clarify this question beyond doubt would require implementing a time-dependent version of the mean-field theory, possibly in linearised form, which I understand has not been done. I leave it optional for the authors to make further comments on this in their manuscript.

Otherwise I am happy with the revisions on the manuscript.

---

## Round 2 · Author Response

Dear Editors,

We are grateful to the Referees for their careful reading of our paper, for their very positive opinion, and for valuable comments and suggestions. Below we give the detailed response to the criticism and the summary of changes. We hope that you find the revised version suitable for publication.

Sincerely,

Jules Givois, Andrea Tononi, and Dmitry Petrov

Point 1 of Referee 3: In section 3 on page 4 a claim is made about the existence of metastable bound states of clusters for finite N_l, e.g. for 1.6 < alpha < 2.3 but it is unclear on which basis this conclusion is drawn as no evidence is presented besides the lines drawn in Fig. 1. Was this conclusion drawn from fully self-consistent numerical calculations, or from the variational analysis? Do we know that these solutions are metastable or might they be (dynamically) unstable solutions of the numerical self-consistent procedure? Why does the corresponding line (e.g. N_l = 2) in Fig. 1 not extend to the full interval of claimed metastability (It looks like the grey dotted line terminates at > 1.8)? Ideally I would like to see a stability analysis of the numerically found non-ground-state solutions, or at least a clarification about what is known about metastability and how the conclusions were reached.

Response:
The bound state of two N+1 clusters does persist down to alpha=1.6. The gray dotted line terminates there. We clarify it in the text of the revised version. That the bound states with N_l=2 and N_l=3 are metastable, i.e., dynamically stable, is the result of an extensive numerical analysis. We have very carefully studied the behavior of these clusters near their respective critical alpha. The distance between the peaks and the energy there have branch-cut singularities as a function of alpha. Although this fact is empirical, it is very well explained by the manner in which the minimum of E(xi) disappears in the variational analysis (see the left panel of Fig.4). We use this branch-cut behavior to numerically determine the critical alpha. We mention these points in the new version.

Point 2 of Referee 3: The beginning of Sec. 3 on page 3 announces three types of solutions, but the third solution is only mentioned two pages later on page 5. To improve the readability, I would suggest to briefly summarise the character of the three solutions at the beginning of the section before discussing them in detail.

Response:
In the revised version we announce these regimes right away.

Point 1 of Referee 2: Please add an appropriate reference to the second sentence of the last paragraph of the introduction ([19]?).

Response:
We insert this reference in the revised version.

Point 2 or Referee 2: The conclusion seems a bit brief... I think it would be worth adding a sentence on your findings for only two N+1 clusters (to complement what you say about a polymer of these clusters in the thermodynamic limit). Also, you mention that the small-N problem should be tackled by other methods. Any suggestions of what a useful approach might be in this case?

Response:
We modify the conclusion section according to these suggestions of the Referee.

---

## Round 2 · List of Changes

1) We cite Ref. [19] in the second sentence of the last paragraph of the introduction.

2) At the beginning of Section 3, we list the three types of solutions before going in more details for clarification.

3) In the second paragraph of Section 3, we add a discussion on the metastable character of the bound-state solution.

4) In the conclusion, we mention our findings for two N+1 clusters and suggest methods, which could be used to solve the quantum problem for finite number of particles.

5) Interchanged Ref. [30] and [31] so that they would be cited in the chronological order.

---

## Editorial Decision

published